# Scattered Train Bolt Point Cloud Segmentation Based on Hierarchical Multi-Scale Feature Learning

**DOI:** 10.3390/s23042019

**Published:** 2023-02-10

**Authors:** Ni Zeng, Jinlong Li, Yu Zhang, Xiaorong Gao, Lin Luo

**Affiliations:** School of Physical Science and Technology, Southwest Jiaotong University, Chengdu 610031, China

**Keywords:** point cloud, deep learning, bolt segmentation, denosing, downsampling

## Abstract

In view of the difficulty of using raw 3D point clouds for component detection in the railway field, this paper designs a point cloud segmentation model based on deep learning together with a point cloud preprocessing mechanism. First, a special preprocessing algorithm is designed to resolve the problems of noise points, acquisition errors, and large data volume in the actual point cloud model of the bolt. The algorithm uses the point cloud adaptive weighted guided filtering for noise smoothing according to the noise characteristics. Then retaining the key points of the point cloud, this algorithm uses the octree to partition the point cloud and carries out iterative farthest point sampling in each partition for obtaining the standard point cloud model. The standard point cloud model is then subjected to hierarchical multi-scale feature extraction to obtain global features, which are combined with local features through a self-attention mechanism, while linear interpolation is used to further expand the perceptual field of local features of the model as a basis for segmentation, and finally the segmentation is completed. Experiments show that the proposed algorithm could deal with the scattered bolt point cloud well, realize the segmentation of train bolt and background, and could achieve high segmentation accuracy, which has important practical significance for train safety detection.

## 1. Introduction

Safety is the main research focus in the railway field [1,2,3]. As vital components affecting the safety of running trains, bolts have always been the key detection units in railways [4]. As shown in Figure 1, bolts play an important role in connecting and fixing various components on the train and are widely used. Traditional detection methods are generally manual troubleshooting of bolt faults by professional train inspectors, but these methods are inefficient, with high costs, human error, and an inability to realize dynamic detection [5]. Therefore, methods using computer vision have been developed. These methods analyze 2D images to judge whether the bolts are missed, damaged, or have other abnormalities, and to improve the bolt detection efficiency and accuracy to some extent [6]. Compared with 2D images, the 3D point cloud contains richer information, such as coordinates [7], which are conducive to the measurement of geometric parameters and has been widely used in many fields, for instance, automatic driving [8], face recognition [9,10], medical diagnosis [11,12], smart cities [13,14], industrial design [15], etc. Up to now, there are a few reports about 3D point clouds detection using the deep learning method in railways. A point cloud identification network based on deep learning is presented in [16], which can be used to identify some components of the railway catenary. However, based on the classification results of a single frame, the network is prone to misclassification in the point cloud connection area, thus causing a negative impact on identification accuracy. Recently, the GTAINet has been proposed to segment the locking wire component [17], which consists of two stages: point cloud classification and point cloud segmentation, but the performance is not satisfactory on scattered point clouds of key components of the train. Obviously, the 3D point cloud has great potential in the field of train safety detection, but it is faced with some problems, such as the unclear boundary of the point cloud connection area of different components, and the poor recognition effect of the scattered point cloud. The bolt, as the train key component, is the focus of detection. Therefore, it is of great research value to realize the recognition and segmentation of bolts in a point cloud of raw train key components collected by laser scanning equipment.

The raw point clouds of train key components used in this paper were collected and provided by inspection personnel at the railway site. According to the analysis of these point clouds, it is found that these point clouds generally have problems such as noise points, blurred boundaries, and a large number of points, which will affect the accuracy of the later task of the point cloud, but also bring large calculation costs. Therefore, an efficient preprocessing framework is particularly important. In this paper, a point cloud adaptive weighted guided filtering (AWGF) algorithm and a fast farthest point sampling algorithm based on key points (FFPS-kp) are designed based on the characteristics of noise and point number. After the point cloud preprocessing is completed, a deep learning network is used to segment the point clouds. First, after hierarchical and multi-scale feature extraction, global features are obtained, and these global features are used twice: (1) input them into the subsequent network directly for upsampling using the inverse distance weighted (IDW) interpolation method; (2) combine with a self-attention mechanism to generate weights of optimization characteristics, and the strengthened or suppressed features are used as the input of the skip connection. The other end of the skip connection is the global features sampled by IDW interpolation. After splicing these two features, they are the final basis for point cloud segmentation.

Our major contributions can be summarized below:

1. A 3D point cloud smoothing and edge-preserving algorithm is introduced which can adjust the filter weight adaptively, which enables us to smooth the scattered point clouds of train key components and get clearer contour lines;

2. A fast farthest point sampling algorithm of key points is proposed, which improves the speed of farthest point sampling (FPS) and saves the calculation cost without sacrificing accuracy;

3. A 3D point cloud segmentation framework suitable for train key components such as bolts is designed, and a deep learning network for bolt point cloud segmentation was trained. The original point cloud with noise points could be processed directly, and a high segmentation accuracy and mIoU were achieved in the dataset of the point cloud of the train bolt components.

The remainder is structured as follows: In Section 2, the related work is introduced. Section 3 describes the overall network architecture and key modules. Section 4 gives the experiment and results analysis respectively. The conclusion is given in Section 5.

## 2. Related Work

Bolt detection based on computer vision technology generally resort to 2D images obtained by cameras. For example, Li et al. trained a support vector machine (SVM) classifier to distinguish bolts from images based on local binary pattern (LBP) descriptors realized positioning by the rotate-and-slide window method and then identified whether the bolts were normal according to whether the bolts had a hexagon shape [4]. In [18], a pioneering vision-based approach was introduced to capture digital images from target bolt connections and estimate bolt rotation and loosening signals through a series of image computing steps. Ref. [19] researched an image-register-based steel node bolt loosening detection method, and the images before and after the bolt are released are registered to detect the differences in the registration error. In [20], by synthesizing 2D images of bolts generated from the graphic model, the deep learning model is trained to achieve bolt recognition and looseness detection, and the applicability of the depth learning model based on vision is improved in practical applications. Overall, the method based on 2D images improved the efficiency and accuracy of bolt detection to a certain extent. But due to the outstanding advantages of the 3D point cloud, bolt detection methods based on the point cloud have broader development prospects, but there are also some difficulties: (1) There are more noise points in the actual collected point cloud of train bolts, and (2) a large amount of point cloud data leads to high computational complexity in late point cloud processing.

First, the problem of acquisition error and noise can be smoothed by the position of the correction points. Ref. [21] proposed a point cloud bilateral filtering (BF) algorithm using position information and normal information. The basic idea is to adjust the position by moving the noise points near the main point cloud along the direction of the normal vector. This method has a large computational overhead and gradient inversion problems. The moving least squares (MLS) method was applied to the problem of point cloud noise [22]. This method iteratively projected the noisy points onto the estimated plane to reconstruct the smooth surface, but with a large amount of calculation, it could not deal with outliers well [23]. In [24], a locally optimal projection operator (LOP) is introduced, but the LOP tendency fails to converge when the point cloud is unevenly distributed. MLS and LOP are also not suitable to deal with the point cloud model of train key components. Point cloud-guided filtering (GF) is proposed in [25], compared with BF, GF has higher efficiency and a better edge-preserving effect, without gradient inversion. However, GF adopts the same smoothing parameters for the whole point cloud and fails to fully consider the details of the point cloud, resulting in a blur in areas with prominent details and obvious edges, and flat areas are overcorrected.

To solve the problem of large amounts of point cloud data, there is a series of point cloud descending sampling algorithms. In deep learning networks, FPS [26] and random sampling (RS) [27] are usually adopted to downsample the point cloud. Among them, the FPS can best retain the contour information of the point cloud and is widely used in the point cloud processing framework based on deep learning, but the computational complexity of FPS is relatively high [28].

At present, the 3D point cloud segmentation algorithms based on deep learning can be roughly divided into direct segmentation methods and indirect segmentation methods [29]. Among them, the former methods include the feature extraction algorithm based on the original points, that is, the original point cloud is segmented directly, and these methods are the mainstream direction of current research [30]. In 2017, the pioneering work that input points into the network to process and extract features directly was proposed and named PointNet by Qi et al. [31]. PointNet introduces the idea of deep learning to directly process the point cloud, and the overall network architecture adopts FPS to complete the downsampling operation and retain the global features of the point cloud through the max-pooling operation. Therefore, the lack of expression of local detail features leads to the poor generalization ability to complex scenes. In the same year, Qi et al. proposed PointNet++ as the optimization of PointNet. To some extent, PointNet++ overcame the shortcoming of lacking local information in PointNet by adopting hierarchical feature learning architecture and sampling layer and by grouping the layer to acquire the overall features of the local area and to improve the performance of point cloud classification and segmentation [32]. Later, PointCNN [33] was proposed, and this network uses 2D convolution on the point cloud to perform semantic segmentation. χ-Conv is used for aggregating spatial structure information and local feature information of points, but the replacement invariance of the point cloud is not considered adequately. In 2019, several networks were proposed to deal with point cloud segmentation tasks. DGCNN [34] extracted local shape features of the point cloud by Edge-Conv and retained the constancy of the arrangement. Semantic information of point sets could be better learned by dynamically updating graph structures between layers, but some local geometric information was still lost. To make the best of geometric information about point cloud shape, Liu et al. proposed RSCNN [35]. The core algorithm is to use a shared multi-layer perceptron (MLP) to obtain the convolution parameters from the geometric topological relations of each point in the point cloud and extract topological constraint relations. PointConv [36] applied a convolutional neural network (CNN) to the 3D point cloud processing task directly, and learned the weight function by using MLP and density function through kernel density estimation, so as to overcome the shortcoming of non-uniform sampling of FPS and realize the invariance of the point cloud order. With the impressive performance of the transformer in the field of natural language analysis and 2D image processing, PCT [37] was proposed, which successfully introduced the attention mechanism and transformer module into the field of 3D point cloud processing, and adopted the self-attention mechanism and offset-attention mechanism for feature learning. It performs well in classification, segmentation, and normal estimation tasks. PointMLP [38] abandons complex feature extractors and uses multiple feedforward residual MLP frameworks to learn point cloud representation, and aggregates local features extracted by MLP for feature learning.

## 3. Network Architecture

In this paper, a 3D point cloud segmentation network is introduced, whose input is some disorderly, scattered, and noisy point clouds. The overall process is shown in Figure 2.

First, the raw point cloud is preprocessed, including denoising and simplification. Adaptive weighted guided filtering of the point cloud (AWGF) is used for correcting the noisy points and smoothing the point cloud. Then, the fast farthest point sampling algorithm of key points (FFPS-kp) is used for simplifying and improving the quality of the point cloud. The preprocessed point cloud model is extracted through multi-scale and multi-layer feature extraction, and the global features are extracted for subsequent point cloud segmentation. The feature extraction layer consists of four parts: sampling layer, grouping layer, convolution layer, and max-pooling layer. The receptive field obtained after multiple feature extraction is large and almost close to the global information. Then, the local features of global regions are expanded by step-by-step upsampling through IDW. At the same time, the global features of local regions extracted from each feature are optimized through the self-attention mechanism, and all features used for point cloud segmentation are obtained by skip connection with the expanded local features of global regions. Then, scores are output through the fully connected network to complete the segmentation task.

### 3.1. Adaptive Weighted Guided Filtering of Point Cloud

Due to the impact of the scanning device, the external environment, and the target object itself, the point cloud obtained directly by the laser scanner will have abnormal data such as noise points or outliers inevitably. These abnormal data will affect the accuracy and efficiency of the subsequent point cloud task, so it is essential to denoise the raw point cloud data [39,40]. The adaptive weighted guided filtering (AWGF) algorithm of the 3D point cloud is shown in Figure 3.

First, the topological relationship is constructed by using K-D tree for the input point cloud $P=\left\{p_{1},p_{2},\ldots ,p_{N}\right\}$, where we use subscripts to represent different points, there are N points in P, and we use $p_{i}$ to represent any one point in P. the parameter k of the search range of each sampling point $p_{i}$ is determined and the neighborhood $N(p_{i})$ is queried, $N(p_{i})$ is the set of points within the k nearest neighbor of $p_{i}$. $p_{ij}$ is used for representing the j-th neighborhood point of $p_{i}$ in $N(p_{i})$, $p_{ij}^{\prime }$ is the point cloud after filtering and smoothing, and $P^{\prime }=\left\{p_{1},p_{2},\ldots ,p_{N}\right\}$ is the output point cloud after the overall filtering and smoothing of the input point cloud P, the number of point clouds after smoothing is the same as the input point cloud, both of which are N. In $N(p_{i})$, the linear transformation model can be formulated as follows:(1)$$p_{ij}^{\prime }=\alpha_{i}p_{ij}+\beta_{i}$$
where, $\alpha_{i}$ and $\beta_{i}$ are linear model parameters limited by the preset neighborhood, which can be calculated by minimizing the cost function J(α,β). The J(α,β) reflects the difference between the output point cloud $P^{\prime }$ and the input point cloud P, the J(α,β) is formally defined as below:(2)$$J(\alpha ,\beta )=\sum_{p_{ij}\in N(p_{i})}\left[{\left(p_{ij}^{\prime }-p_{ij}\right)}^{2}+\varepsilon \alpha_{i}^{2}\right]$$
where, ε is the parameter used for controlling the smoothness. In order to better handle edges, an edge perception weight $w_{\left(p_{ij}\right)}$ is set as:(3)$$w_{\left(p_{ij}\right)}=\frac{1}{\left|N(p_{i})\right|}\sum_{p_{ij}\in N(p_{i})}^{}\frac{\sigma_{i}^{2}\left(p_{ij}^{}\right)+\lambda }{\sigma_{i}^{2}\left(p_{i}\right)+\lambda }$$
where, $\sigma_{i}^{2}\left(p_{ij}^{}\right)$ is the variance of the distance between $p_{i}$ and the surrounding j neighborhood points in $N(p_{i})$, and $\sigma_{i}^{2}\left(p_{i}\right)$ is the variance of the distance between all points in $N(p_{i})$ and their neighborhood points, λ is a constant. The edge perception weight $w_{\left(p_{ij}\right)}$ is used for assigning more weight to the points at the edge than the points in the flat region. Accordingly, the regularization coefficient at the edge is smaller, so the edge could be well preserved, and the position of noise points with a long distance could be adjusted. Therefore, the cost function is transformed below:(4)$$J(\alpha ,\beta )=\sum_{p_{ij}\in N(p_{i})}^{}[{\left(\alpha_{i}p_{ij}+\beta_{i}-p_{ij}\right)}^{2}+\frac{\varepsilon }{w_{\left(p_{ij}\right)}}\alpha_{i}^{2}]$$

When the cost function J(α,β) is the minimum value, the values of $\frac{\partial J(\alpha ,\beta )}{\partial \alpha_{i}}$ and $\frac{\partial J(\alpha ,\beta )}{\partial \beta_{i}}$ are equal to 0. Therefore, the linear model parameter solutions could be calculated as Formulas (5) and (6):(5)$$\alpha_{i}=\frac{⟮\frac{1}{\left|N(p_{i})\right|}\sum_{{}_{p_{ij}\in N(p_{i})}}^{}p_{ij}.p_{ij}-\bar{p}.\bar{p}⟯}{\left[⟮\frac{1}{\left|N(p_{i})\right|}\sum_{{}_{p_{ij}\in N(p_{i})}}^{}p_{ij}.p_{ij}-\bar{p}.\bar{p}⟯+\frac{\varepsilon }{w_{\left(p_{ij}\right)}}\right]}$$
(6)$$\beta_{i}=\bar{p}-\alpha_{i}.\bar{p}$$
where
(7)$$\bar{p}=\frac{1}{\left|N(p_{i})\right|}\sum_{p_{ij}\in N(p_{i})}p_{ij}$$

### 3.2. Fast Farthest Point Sampling Based on Key Points

Since the number of the 3D point cloud obtained by scanners is huge, such large point cloud data entering the deep learning network directly will make the training time too long. In another word, if there is a downsampling process, the network efficiency will be improved. The downsampling algorithm named FFPS-kp is shown in Figure 4.

The denoised point cloud $P^{\prime }=\left\{p_{1}^{\prime },p_{2}^{\prime },\ldots ,p_{N}^{\prime }\right\}$ is sampled by FFPS-kp. FFPS-kp consists of the following key steps: First, select a point in $P^{\prime }$, which is $p_{i}^{\prime }$ and create a K-D tree with $p_{i}^{\prime }$ in the point cloud $P^{\prime }$ as the centroid, and the k nearest neighbor (KNN) algorithm is used for selecting k points around $p_{i}^{\prime }$ as the neighborhood; $p_{ij}^{\prime }$ is a neighborhood point of $p_{i}^{\prime }$, the j-th point in the KNN is represented by the subscript j, and the covariance matrix $\mathrm{cov}\left(p_{i}^{\prime }\right)$ is constructed for the neighborhood of $p_{i}^{\prime }$. The covariance matrix $\mathrm{cov}\left(p_{i}^{\prime }\right)$ can be formulated as follows:(8)$$\mathrm{cov}\left(p_{i}^{\prime }\right)=\frac{1}{k}{\sum_{p_{ij}^{\prime }\in N(p_{i}^{\prime })}\left(p_{ij}^{\prime }-p_{i}^{\prime }\right)}^{T}\left(p_{ij}^{\prime }-p_{i}^{\prime }\right)$$

The three feature vectors of $\mathrm{cov}\left(p_{i}^{\prime }\right)$ form an orthogonal basis (a,b,c), and the local reference frame is established with $p_{i}^{\prime }$ as the coordinate origin of local reference frame and (a,b,c) as the coordinate axis. The three eigenvalues $\gamma_{1},\gamma_{2},\gamma_{3}$ of $\mathrm{cov}\left(p_{i}^{\prime }\right)$ could approximately represent the complexity of the surface at $p_{i}^{\prime }$, and the curvature is defined as:(9)$$c_{i}=\frac{\gamma_{3}}{\gamma_{1}+\gamma_{2}+\gamma_{3}}$$

After the curvature of all points in $p_{i}^{\prime }$ are obtained, they are sorted in an increasing sequence and the first $n_{1}$ points are taken out as the feature points ($n_{1}$ can be adjusted according to the actual demand). Then the octree for the remaining point cloud is established and partitioned [41]. In each region, FFPS-kp is carried out, and the number of sampling points is taken as $n_{2}=n-n_{1}$. The process is as follows: first, the x,y,z coordinates of each point of the input point cloud are read, and the minimum envelope cubes of the point cloud are obtained; then it is bisected in the x,y,z directions, and thus it is divided into eight subcubes with a serial number. Then, given an index, the number of points in each subcube were counted. Meanwhile, the ratio of the number of points in each subcube to the total number of the point clouds was calculated as the ratio of the number of the sampling points in this subcube to the total sampling points, to complete the distribution of the sampling points. Next, an initial point is selected in each subcube, and the distances between the remaining points to the selected point are calculated, the point with the maximum distance is taken out, and this process is repeated until the target points are taken out. If the actual sampling points are more than those needed, the extra points are randomly removed from the subcube with the most sampling points; otherwise, the corresponding points are randomly selected from the region with the most sampling points to complete the partition sampling.

Finally, the combination of curvature sampling and partition FFPS-kp is regarded as the final sampling result, set as $P^{\prime \prime }=\left\{p_{1}^{\prime \prime },p_{2}^{\prime \prime },\ldots ,p_{n}^{\prime \prime }\right\}$, where n is the number of points after downsampling and n<N, which is input into the subsequent network for training.

### 3.3. Feature Extraction

To improve the segmentation accuracy, both local features and global features should be considered when feature extraction is carried out. For example, a fatal weakness of PointNet is that it fails to preserve the local features, resulting in lack of detailed description capabilities [31]. For this problem, both the global and local feature descriptions are fully considered in the feature extraction network, in which a multi-scale and hierarchical feature extraction module combing attention mechanism is introduced. The overall feature learning process is shown in Figure 5.

The global feature learning process is as follows:

(1) The preprocessed point cloud $P^{\prime \prime }$ is input into the feature extraction network;

(2) FFPS-kp is used for sampling the $P_{1}^{\prime \prime }=\left\{p_{1}^{\prime \prime },p_{2}^{\prime \prime },\ldots ,p_{n}^{\prime \prime }\right\}$, and $P_{1}^{\prime \prime }$ is set as the center of a sphere to determine the neighborhood radius $r_{1}$ and delimit the neighborhood sphere $s_{1}$;

(3) MLP and max-pooling are used for extracting and concentrating the features of all points in $s_{1}$, and the $F_{1}=\left\{f_{1},f_{2},\ldots ,f_{m_{1}}\right\}$ after the first feature extraction is obtained;

(4) For $F_{1}$, the second feature extraction is carried out in the same way. The neighborhood radius $r_{2}$ and the neighborhood sphere $s_{2}$ are set to obtain the $F_{2}=\left\{f_{1},f_{2},\ldots ,f_{m_{2}}\right\}$; 

(5) The third feature extraction is carried out for $F_{2}$, and the neighborhood radius $r_{3}$ and the neighborhood sphere $s_{3}$ are set to obtain the $F_{3}=\left\{f_{1},f_{2},\ldots ,f_{m_{3}}\right\}$, where $r_{1}<r_{2}<r_{3}$, $m_{3}<m_{2}<m_{1}$;

(6) Finally, the $F_{3}$ with global features is output to prepare for the subsequent point cloud segmentation.

The $F_{3}$ extracted with hierarchical clustering and multi-scale features is enough to describe the global features, but it still lacks detailed features when used for segmentation. In order to consider the description ability of the detailed information, features are expanded, including feature interpolation and feature splicing. First, based on $F_{3}$, the IDW interpolation method based on the KNN is used for upsampling to achieve the purpose of supplementing details. The basic principle of IDW interpolation is that points that are closer to each other are more similar than those farther apart. 

However, the point cloud features obtained by upsampling fail to make the best of the point cloud local information obtained during feature extraction. Therefore, the global features of local regions extracted by hierarchical multi-scale features are spliced into the corresponding global feature layer after upsampling using skip connection. In the process of skip connection, the self-attention mechanism is adopted for feature optimization of the global features in local regions, and then it is combined with the global features, so that the global features can better express the local information of the point cloud and also improve the feature expression ability and the segmentation accuracy accordingly. The core of the self-attention mechanism is to use other information in the target to enhance the semantic representation of the target information and make better use of the context of the target information [42].

## 4. Experiments and Discussion

### 4.1. Experimental Environment and Parameters

The experimental hardware configuration is NVIDIA GeForce RTX 3060, and the software environment is Windows 10, Python3.6.12, PyTorch 1.7.1, CUDA 11.2. The experimental parameters were set as Batch size 24 and Epoch 200. An Adam optimizer was used when training networks. The initial learning rate is set to 0.001, the learning rate decay index is 0.0001, and the activation function is RELU; the loss function is the cross entropy for the segmentation network.

### 4.2. Dataset

In this experiment, a laser 3D scanner was used to collect the data of point clouds containing bolts as train key components. We selected 1256 point clouds containing bolts and point cloud labels were assigned. Label 0 represented bolts and label 1 represented the background. Each point cloud in the datasets containing the bolt position consists of its X, Y, Z coordinates, X, Y, Z normal values, and labels. Table 1 shows the dataset details.

### 4.3. Evaluation Metrics

In this paper, time is used to evaluate the calculated efficiency of denoising and sampling, overall accuracy (OA). Intersection over union (IoU), and mean intersection over union (mIoU) are used for evaluating the segmentation effect. The calculation formulas are listed below:(10)$$\mathrm{OA}=\frac{TP+TN}{TP+FP+TN+FN}$$
(11)$$IoU=\frac{TP}{TP+FP+FN}$$
(12)$$mIoU=\frac{1}{k}\sum IoU$$

### 4.4. Results and Discussion

#### 4.4.1. Denoising and Smoothing Experiment

In order to compare the performance of the denoising and smoothing algorithm AWGF in this paper with the BF algorithm and the GF algorithm, a comparative experiment was conducted on four bolts in the dataset. The corresponding number of bolts is shown in Table 2, and the time comparison experiment is shown in Table 3. Figure 6 shows the contrast details.

Through the analysis of Table 2 and Table 3, in terms of computational complexity, with the increase of the number of point clouds, the filtering time of the three smoothing algorithms increases correspondingly, among which the BF takes the longest time, almost twice that of GF and AWGF. The GF gives up the calculation of the point cloud normal vector and uses the linear model, which saves the calculation cost greatly. Due to the introduction of edge perception weight calculation, the AWGF brings a certain calculation overhead, resulting in a slight increase in the filtering time of the AWGF compared with the GF.

Figure 6 shows the comparison between the raw bolt point cloud and the point cloud smoothed by the BF, the GF, and the AWGF, respectively. By comparing the experimental results, it is obvious that the three algorithms could achieve a certain smoothing effect. As can be seen from the details in Figure 6b, the model filtered by the BF algorithm still contains some noise points, and the smoothness is barely satisfactory. Figure 6c is smoother than that in Figure 6b. It can be seen the GF has a better smoothing effect on noise than the BF algorithm and can realize the smoothing task of the bolt point cloud, making the boundary contour of the point cloud clearer. Figure 6d shows the experimental results of the algorithm we proposed. Compared with Figure 6c, the filtering effect is more satisfactory and the correction effect on noisy points is better, which proves the effectiveness of the edge perception weight operator in this paper.

To sum up, by analyzing the experimental data and comparing the smoothing results in Figure 6, we can be conclude that the AWGF can achieve a balance between computational performance and filtering performance, and better complete the denoising and smoothing tasks of the bolt point cloud.

#### 4.4.2. Downsampling Experiment

In order to compare the performance of the proposed algorithm FFPS-kp with FPS, four bolt point clouds were selected from the datasets for the downsampling comparison experiment, and the sampling numbers were set as 512, 1024, and 2048, respectively. The number of sampling points and the corresponding sampling time are listed in Table 4 and Table 5; the effect after sampling is shown in Figure 7.

Table 4 and Table 5 show the time spent using FPS and FFPS-kp for different point numbers of the same point cloud and different point cloud models, respectively. When sampling different points in the same point cloud, the sampling time of FPS increases rapidly with the increase of sampling points, and the average time is about 20–30 times that of FFPS-kp. For a large number of points of the train bolt point cloud, FPS can complete the sampling, but the time is too long. However, the FFPS-kp algorithm has an obvious speed advantage for models with more points, which can quickly complete the sampling task and greatly improve the sampling efficiency.

Figure 7 shows that both FPS and FFPS-kp can complete the downsampling of bolt point cloud components, retaining surface boundary and contour information fully. While compared with FPS, FFPS-kp can preserve the key points as much as possible, for instance, for the areas with large curvature, and for those with obvious surface characteristics such as the junction of bolts and background. These key points can provide some help for the later point cloud segmentation task, which is also proved in the following segmentation experiments.

#### 4.4.3. Bolt Segmentation

In order to evaluate the performance of the proposed network architecture on the bolted point cloud segmentation task, we selected several popular point cloud segmentation algorithms, PointNet, Pointnet++, PointConv, DGCNN, and PointMLP to make the comparison. To reduce the calculation consumption in the preprocessing part of the experiment, the actual dense bolt point cloud was downsampled, and 2048 points were reserved for feature extraction. Table 6 shows the comparison of segmentation accuracy, IoU, and mIoU. Figure 8 gives the comparative graph of segmentation results of different deep learning algorithms.

Table 6 shows the segmentation performance comparison between the segmentation framework in this paper and the popular point cloud segmentation networks on the bolt point cloud datasets. As can be seen from Table 6, the point cloud segmentation network we proposed achieved the best segmentation performance, which are 98.12% OA, 95.18% mIOU, 94.54% bolt IoU, and 95.81% background IoU, respectively. In terms of the mIoU, compared with PointNet, Pointnet++, PointConv, DGCNN, and PointMLP, the mIOU increased by 4.94%, 3.19%, 1.95%, 2.09%, and 0.69%, respectively. In terms of the OA, compared with them, the OA increased by 2.85%, 1.80%, 1.02%, 1.43%, and 0.43%, respectively.

To compare the proposed method with other methods visualized, we selected some bolts randomly and compared the segmentation effect in Figure 8. The colors used refer to Wang et al. [17], where the green color denotes the bolt and the red color denotes the background.

As shown in Figure 8, although most of the segmentation algorithms have achieved a good segmentation effect, some algorithms cannot handle details well, such as the connection between bolts with background as well as the connection between bolts. However, the edge smoothing algorithm we proposed in this paper makes the connection between the bolt and the background smoother, that is, the boundary between the two is clearer, resulting in better overall segmentation effect and better performance in details.

#### 4.4.4. Ablation Experiment

In order to explore the influence of the denoising module, the sampling module, and the attention mechanism module on the bolt segmentation results, we conducted a comparison of module ablation, as shown in Table 7.

Table 7 lists the comparison of the results of the ablation experiments of these modules. To explore whether the preprocessing module contributes to the segmentation results, we kept the feature extraction part unchanged and selected different denoising methods and different downsampling methods for experiments. According to the experimental results, when the denoising methods are the same, that is, when AWGF is used for denoising, FFPS-kp as the downsampling method, we obtained higher mIoU and OA. Compared with FPS, mIoU is higher by 0.08% and OA is higher by 0.12%. In addition, the sampling method is the same, that is, when FFPS-kp is used for sampling, the segmentation effect of AWGF is better than that of GF, mIoU is increased by 0.51%, and OA is increased by 0.23%. To verify the effectiveness of the attention module, we maintained the same preprocessing module and used or deleted the attention module to carry out the comparative experiment. When AWGF is used as the point cloud denoising algorithm and FFPS-kp is used as the downsampling algorithm, the addition of the attention mechanism module increases mIoU by 0.78% and OA by 0.87%, which verified the effectiveness of self-attention mechanism module in the process of bolt point cloud segmentation.

In short, the ablation experiment verifies the effectiveness of the proposed preprocessing module in point cloud segmentation, including the denoising module and the downsampling module. It also shows that the self-attention mechanism can make better use of local information, strengthen the extraction ability, and improve the segmentation performance.

## 5. Conclusions

Aiming at the actual train bolt point cloud model, we established a hierarchical and multi-scale feature extraction point cloud segmentation algorithm, which includes filtering smoothing, downsampling, and other preprocessing operations combined with an attention mechanism. First, the AWGF algorithm proposed in this paper can remove the noise points in the scattered point cloud, making the contour of the point cloud smoother and the boundary clearer. To solve the problem of downsampling, we optimized the FPS algorithm and proposed FFPS-kp. Compared with FPS, FFPS-kp greatly shortens the sampling time and improves the sampling efficiency without losing the segmentation accuracy. Finally, the hierarchical multi-scale point cloud segmentation model proposed in this paper performs well in the segmentation experiment of the actual train bolt point clouds and their backgrounds, and can successfully segment the bolt point clouds, which can be used in actual railway environments.

However, this algorithm takes too long time to segment the point cloud in large environments. In the future, we would consider designing a more efficient point cloud downsampling algorithm that could pay more attention to key features, so as to reduce the calculation amount for later feature learning, such as the idea of the critical points layer. At the same time, we would try to simplify the feature extraction structure, and further shorten the time without losing the accuracy and mIoU, so that the network could be applied to the rapid segmentation of the original point cloud in large train environments.

## Figures and Tables

**Figure 1 sensors-23-02019-f001:**
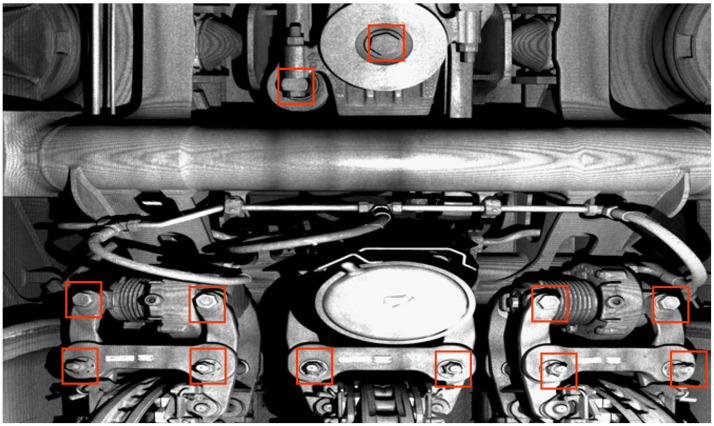
Scenes from an underneath high-speed train. The red square marks the bolts.

**Figure 2 sensors-23-02019-f002:**
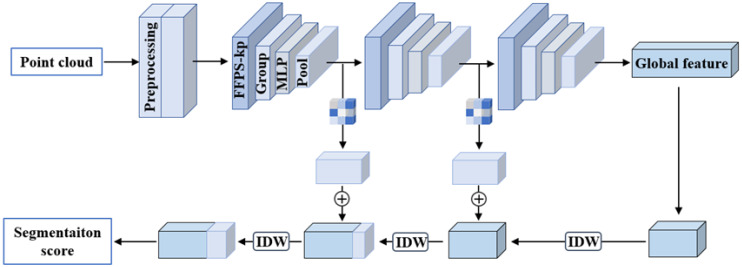
Architectures of the segmentation network.

**Figure 3 sensors-23-02019-f003:**
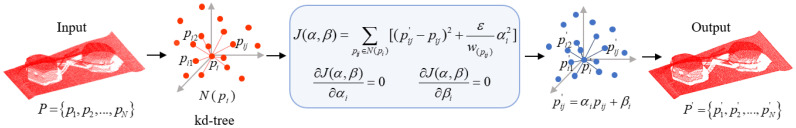
The pipeline of the AWGF.

**Figure 4 sensors-23-02019-f004:**
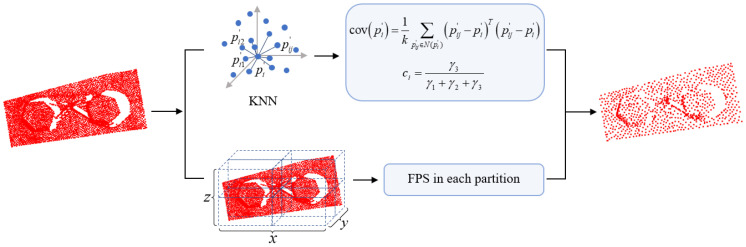
The pipeline of the FFPS-kp.

**Figure 5 sensors-23-02019-f005:**
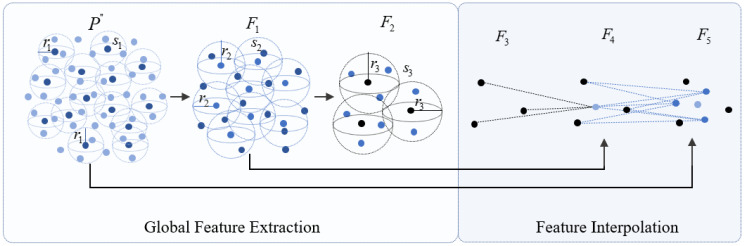
The process of feature learning.

**Figure 6 sensors-23-02019-f006:**
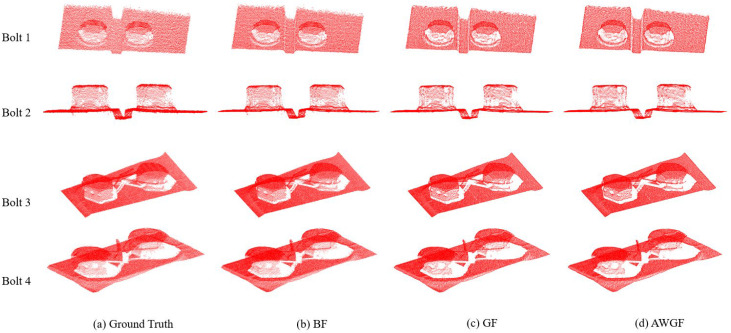
Comparison results of the smoothing.

**Figure 7 sensors-23-02019-f007:**
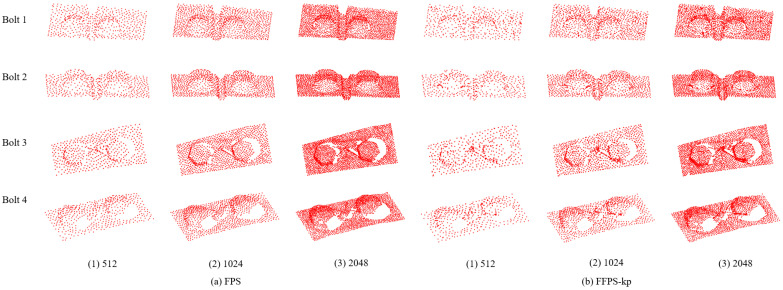
Comparison results of the sampling.

**Figure 8 sensors-23-02019-f008:**
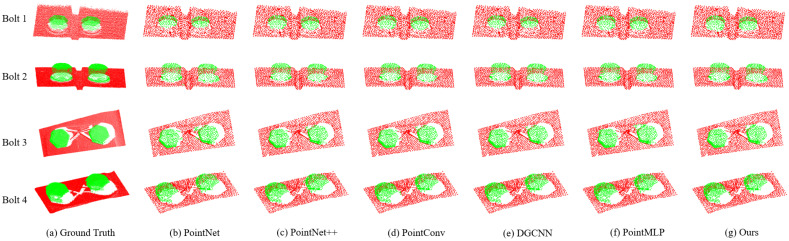
Comparison results of the segmentation.

**Table 1 sensors-23-02019-t001:** Dataset details.

Training	Test	Validation	Total
870	252	134	1256

**Table 2 sensors-23-02019-t002:** Bolts and its number of points.

Point Cloud	Bolt1	Bolt2	Bolt3	Bolt4
Number	35,006	44,316	59,267	62,405

**Table 3 sensors-23-02019-t003:** Smoothing time comparison.

Denoising Method	Bolt1 (s)	Bolt2 (s)	Bolt3 (s)	Bolt4 (s)
BF	0.369	0.413	0.634	0.649
GF	0.180	0.226	0.321	0.341
AWGF	0.215	0.266	0.376	0.395

**Table 4 sensors-23-02019-t004:** FPS sampling time.

Sampling Number	Bolt1 (s)	Bolt2 (s)	Bolt3 (s)	Bolt4 (s)
512	62.43	78.08	103.14	110.84
1024	237.56	279.42	361.42	376.5
2048	989.01	1188.53	1648.36	1640.72

**Table 5 sensors-23-02019-t005:** FFPS-kp sampling time.

Sampling Number	Bolt1 (s)	Bolt2 (s)	Bolt3 (s)	Bolt4 (s)
512	2.72	3.56	5.57	6.23
1024	9.16	12.02	15.64	18.26
2048	35.59	45.34	61.89	70.89

**Table 6 sensors-23-02019-t006:** Comparison of segmentation results.

Method	Bolt (%)	Background (%)	mIOU (%)	OA (%)
PointNet	89.46	91.01	90.24	95.27
PointNet++	91.25	92.72	91.99	96.32
PointConv	92.83	93.63	93.23	97.10
DGCNN	92.65	93.52	93.09	96.69
PointMLP	93.78	95.19	94.49	97.69
Ours	94.54	95.81	95.18	98.12

**Table 7 sensors-23-02019-t007:** Ablation experiment.

Denoising Method	Sampling Method	Attention Mechanism	Bolt (%)	Background (%)	mIOU (%)	OA (%)
AWGF	FFPS-kp	-	93.76	95.04	94.40	97.25
AWGF	FFPS-kp	Self-attention	94.54	95.81	95.18	98.12
AWGF	FPS	Self-attention	94.43	95.77	95.10	98.00
GF	FFPS-kp	Self-attention	94.05	95.13	94.59	97.77

## Data Availability

Not applicable.

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
