# Peer review of "Scattered Train Bolt Point Cloud Segmentation Based on Hierarchical Multi-Scale Feature Learning"

_sensors, 2023, doi:10.3390/s23042019_

Round 1
Reviewer 1 Report
Scattered train bolt point cloud segmentation based on hierarchical multi-scale feature learning is investigated in this manuscript. The paper is well organized but there are several comments should be addressed.
1) The introduction section is too short. The literature review is not enough. Section 2 is suggested to be combined with Section 1.
2)The novelty points should be clearly explained and stressed. If a method is proposed, what are the advantages over other approaches.
3) A comparison with some published methods should be presented, especially for the deep learning.
4) Many symbols in the manuscript are lack of explanations.
5) Please revise the conclusion section. It should focus on the major conclusions instead of repeating what have done in the paper.
Reviewer 2 Report
1) The database used is not very clear regarding.
2) Detailed suggestions for future studies should be added to the Conclusions.
3) Is it possible to improve the results in further research?
4) Authors should improve the state of the art and research gap. It is recommended that authors expand the literature review.
5) Minor rectification needed on grammar and spellings.
Reviewer 3 Report
The authors propose a manuscript titled Scattered train bolt point cloud segmentation based on hierarchical multi-scale feature learning dealing with point cloud segmentation model for train bolt segmentation. This has implications in train safety estimation and thus the significance of the topic is established. The manuscript is interesting and worth publishing however some minor issues must be resolved.
1) Introductory section gives quite useful information about processing of 3D point clouds, however there is no information about how does one acquire 3D points. Please add this information.
2) In Equation 3) add explanation for the variance calculation? is it sum of squared errors betwen pi in pij in N(pi)?
3) Can you add time comparisons between your proposal and other networks that you used for comparison along with other segmentation results in Table 6.?
Round 2
Reviewer 1 Report
The authors did not address my comments well. Some problems are still presented, which are
(1) In the introduction section, the current challenges or shortcoming of current methods should be concluded by literature review. And then a novel method is proposed to solve those problems. Therefore, the literature review is not enough in the Introduction.
(2) some superscript or subscript are misleading, and it is suggested to explain clearly or change by using other symbols.
(3) In the abstract, ‘Then retains the key points of the point cloud’ should be ‘Then retaining…’.
